# Did Children in Single-Parent Households Have a Higher Probability of Emotional Instability during the COVID-19 Pandemic? A Nationwide Cross-Sectional Study in Japan

**DOI:** 10.3390/ijerph19074239

**Published:** 2022-04-01

**Authors:** Takuto Naito, Yasutake Tomata, Tatsui Otsuka, Kanami Tsuno, Takahiro Tabuchi

**Affiliations:** 1Department of Global Public Health, Karolinska Institutet, 171 77 Stockholm, Sweden; takuto.naito@stud.ki.se; 2School of Nutrition and Dietetics, Faculty of Health and Social Services, Kanagawa University of Human Services, Yokosuka 238-8522, Japan; 3Department of Medical Epidemiology and Biostatistics, Karolinska Institutet, 171 77 Stockholm, Sweden; 4Department of Psychiatry, Tohoku University Graduate School of Medicine, Sendai 980-8574, Japan; otsuka@med.tohoku.ac.jp; 5School of Health Innovation, Kanagawa University of Human Services, Kawasaki 210-0821, Japan; k.tsuno-wm4@kuhs.ac.jp; 6Cancer Control Center, Osaka International Cancer Institute, Osaka 541-8567, Japan; tabuti-ta@mc.pref.osaka.jp

**Keywords:** single-parent, COVID-19, children, mental health, Japan

## Abstract

The influence of public health measures against COVID-19 in Japan on child mental health by household type is unknown. This study aimed to investigate whether COVID-19 and the declaration of a state of emergency in Japan affected children’s mental health between single-parent and two-parent households disproportionately. A large cross-sectional online survey was conducted from August to September 2020. The study included 3365 parents with children aged 0–14 years old who reported their children’s mental status during the declared state of emergency. Emotional instability was reported dichotomously by parents. As the primary result, the probability of emotional instability was higher in single-parent households compared with that in two-parent households after adjustments for potential covariates; the adjusted prevalence ratio (95% CI) was 1.26 (1.07–1.49). Our findings suggest a disproportionate impact on children’s mental health due to the pandemic.

## 1. Introduction

Since December 2019, coronavirus disease 2019 (COVID-19) has brought about disastrous impacts globally [1]. Several public health measures to deter the pandemic, including lockdowns, mass screening and contact tracing, have been adopted since early 2020 [2]. The lockdown and social distancing measures substantially affected the lives of both adults and children. According to UNESCO, more than 80% of students worldwide experienced school closures in April 2020 [3].

As in other countries and regions, a rapid increase in the number of COVID-19 cases urged the Japanese government to request that all elementary schools, junior high schools, high schools and special needs schools close from 2 March 2020, until the end of March, 2020 [4,5]. On the basis of the request, 99% of the schools in Japan closed until the beginning of April 2020 [6]. However, the government declared a state of emergency in seven populated prefectures, such as Tokyo, on 7 April 2020. The government, in turn, requested schools in the prefectures affected to close again [7]. Subsequently, the declaration of a state of emergency was announced on 16 April 2020. Therefore, all elementary schools, junior high schools, high schools and special needs schools were again encouraged to close [8]. The declaration was lifted on 25 May 2020. Virtually, many students experienced almost a 3-month school closure.

During the school closure, many schools continued to offer learning opportunities for children. However, online classes were not common in the Japanese compulsory educational system before the COVID-19 pandemic. Hence, the online schooling was rather Emergency Remote Teaching (ERT) than planned online education. ERT is defined as “a temporary shift of instructional delivery to an alternate delivery mode due to crisis circumstances” [9]. In Japan, a local educational committee settled in each municipality is responsible for education provision at municipality-owned schools, which account for most of the public education in Japan, and educational corporations are responsible at private schools. A nationwide survey conducted by the Ministry of Education, Culture, Sports, Science and Technology on 16 April 2020, revealed that 60 of 1213 local educational committees (5%) provided real-time interactive education via online platforms, 118 (10%) developed original lecture videos, 353 (29%) took advantage of other digital textbooks and materials and 288 (24%) utilized television programs [10].

Children in single-parent households in Japan faced challenges not only economically but also physically or mentally even before the COVID-19 pandemic. Single mothers in Japan more frequently reported severe living conditions and psychological distress than two-parent mothers [11]. Japanese adolescents in single-parent households are more likely to experience psychological distress than two-parent households [12].

Furthermore, the pandemic possibly affected single-parent and two-parent households disproportionately [13]. A previous study in Hong Kong suggests that single-parent families are more prone to being adversely affected by COVID-19 and related social restrictions [14]. However, to the best of our knowledge, no study examined the influence of the COVID-19 pandemic on children’s mental health in Japan by household type.

This study aimed to investigate whether COVID-19 and the first declaration of a state of emergency in Japan affected children’s mental health by comparing single-parent and two-parent households. Second, we explored environmental and behavioral factors associated with children’s emotional instability.

## 2. Materials and Methods

### 2.1. Study Design

This cross-sectional study used data on children aged 0–14 years old (information provided by their parents) from the Japan “COVID-19 and Society” Internet Survey (JACSIS) study. The JACSIS study is based on a self-administered web questionnaire survey distributed through a commercial research agency ((Rakuten Insight, Inc., Tokyo, Japan), which had approximately 2.2 million qualified panelists in 2019). Invitation to the survey was sent via e-mail to the panelists. The invitation and the questionnaire were only in Japanese.

The questionnaires were distributed to 224,389 panelists selected by sex, age and residential area (47 prefectures in Japan) based on the random sampling method. Japanese people who lived in other countries were not allowed to participate. The panelists who agreed to answer the survey accessed the designated website and responded to the questions. Options were available to skip some questions or discontinue participation during the survey. Recruitment was started on 25 August 2020 and was completed on 30 September 2020.

When the respondents had multiple children, we asked them to answer the status of only one child among them. The priority among target children was allocated as follows (highest to lowest in order): a child at a higher grade at primary school (10–12 years old); a child at a lower grade at primary school (6–9 years old); a child at junior high school (13–14 years old); and a child before primary school (0–5 years old). With the structural constraint of the web survey, we were restricted to inquiring about only one of the participants’ children. As the JACSIS survey collectively included many questions related to the parent–child relationship, we prioritized the age groups that could be more profoundly influenced by the parent–child relationship. Our assumption was that children younger than teenagers, namely, younger than 12 years old (i.e., younger than the age of junior high school students) would be more substantially affected than teenagers because teenagers are more likely to be independent from their parents. On the other hand, it would be presumably challenging to assess emotional and behavioral changes among infants and pre-school children; therefore, we assigned the lowest priority to infants and pre-school children. In addition to these assumptions, we supposed children of 10–12 years old were likely to independently participate in more extra-curricular activities and leisure activities than 6–9-year-old children. Accordingly, 10–12-year-olds were the ideal age group to be analyzed, and we prioritized this group first.

### 2.2. Study Population

Among the recruited respondents, we excluded individuals who reported unnatural or inconsistent responses using an algorithm shared in JACSIS studies [15]. We excluded respondents who reported that they had no child, or all their children were over 14 years old. Then, we excluded outliers who were younger than the age at which people are legally allowed to marry in Japan, that is, 18 years old for men and 16 years old for women. Furthermore, respondents with two or more children of the same age were excluded because we could not identify which child the participants answered for in the questions of interest. Moreover, we excluded respondents who reported that they had more than five children because the maximum number of children that the respondents could in the survey was five due to the survey structure as they might have failed to provide the appropriate outcome information about children in the prioritized age groups. Lastly, we excluded respondents who did not provide valid information about the primary outcome variable (i.e., they responded “do not know” or “not relevant” to the statement “my child became emotionally instable”).

### 2.3. Outcome (Emotional Instability)

As the primary outcome variable, we asked the parents about their children’s emotional instability during the two months from April to May 2020 (i.e., the period of the first state of emergency in Japan, when schools, nurseries and kindergartens were requested to close). The statement “my child became emotionally instable” was responded to with any one of “yes”, “no”, “do not know”, or “not relevant”. “Yes” was defined as the presence of emotional instability, while “no” was defined as no emotional instability.

As secondary outcome measures, we measured child misbehaviors that can be deemed consequences of children’s emotional instability. Child misbehaviors included violence, abusive language towards others, demotivation to study and school absenteeism during the same two months. We asked the parents whether they agreed with the following statements: “my child perpetrated violence”, “my child spoke abusive words”, “my child was demotivated to study” and “my child stopped going to school”; the parents responded any one of “yes”, “no”, “do not know”, or “not relevant”. “Yes” was defined as the perpetration of the behaviors, while “no” was defined as no perpetration.

### 2.4. Exposure (Single-Parent)

The primary exposure variable was the family type defined by self-reported number of spouses (0 or 1). Respondents who answered no spouse were defined as “single-parent households”, and the other respondents were defined as “two-parent households”.

Children’s environment and extracurricular activities included school and nursery closures, online education in April and May 2020, childcare by others (i.e., relatives and childcare services) due to school and nursery closure, and extracurricular activity participation (i.e., cram schools and sport/hobby lessons), which were reported by either “yes” or “no”.

The daily activities of their children were reported in the following categories: sleeping, studying, physical activity, reading, watching TV and online entertainment and playing games. We asked the parents to select one of the following alternatives that specify the hours spent per day carrying out each of the abovementioned activities: 0, less than 30 min, about 30 min, 1 h, 2 h, 3 h, 4–5 h, 6–7 h, 8–9 h, 10–11 h, ≥12 h and do not know. We then described the proportion of the children in the following groups: 8 h or longer for sleep, 1 h or longer for study, 0.5 h or longer for physical activity, 0.5 h or longer for reading, 2 h or longer for watching TV and online entertainment and 1 h or longer for playing games.

### 2.5. Covariates

We included the following covariates: parents’ age, socioeconomic status and whether they lived with a grandparent. As a socioeconomic status, we included: educational attainment (graduated from college or higher institutions and high school or lower institutions); household income level (categorized using the tertiles of household equivalent income (“low” = less than 2.5 million JPY; “medium” = 2.5 to 4.3 million JPY; and “high” = more than 4.3 million JPY), and “not responded”); the number of family members (≤3, 4 and ≥5); and employment status (employer, self-employed, regular employee, non-regular employee, and unemployed). The household equivalized income was calculated as the self-reported gross (pre-tax) income in 2019, divided by the square root of the number of household members. Whether they lived with a grandparent was dichotomously categorized (yes/no) based on the self-reported number of grandparents living with them.

### 2.6. Statistical Analysis

We performed a descriptive analysis of covariates according to family type. We used Chi-square tests to test the differences in categorical variables and the Mann–Whitney test for parents’ age because the distribution of parents’ age was not assumed as normal distribution.

We calculated the prevalence ratio and 95% confidence interval (95% CI) of emotional instability using Poisson regression with robust variance to examine the association between the family type and children’s emotional instability [16]. Adjustment items for multivariable logistic regression models were respondents’ age groups (15–19, 20–29, … and 70–79), educational attainment, household income, employment status and whether they lived with a grandparent. For two-parent families, as one of the two parents reported all the information, we adjusted for the covariates based on the information from the responding parent only. For sensitivity analyses, we conducted stratified analyses of the following items: children’s age, children’s environment (i.e., school/nursery closure, online education and childcare by others), extracurricular activities (i.e., cram schools and sports/hobby lessons) and daily activities (i.e., sleep ≥8 h, study ≥1 h, physical activity ≥30 min, reading ≥30 min, watching TV and online entertainment ≥2 h and playing games ≥1 h). As for the stratified analyses, we excluded the respondents who did not provide valid information necessary for stratification as we could not classify them into either of the two groups. In addition, to test the interaction effects, we performed adjusted generalized linear model analyses with an interaction term of stratification variables (i.e., the child age category x child environment or child extracurricular activities).

We performed the analyses using STATA version 16.1. We considered two-sided *p*-values < 0.05 as statistically significant.

## 3. Results

### 3.1. Characteristics

During the study period, 29,000 respondents (response rate 12.9%) answered the questionnaires. We excluded 2601 individuals who reported unnatural or inconsistent responses using an algorithm we developed, which left 26,399 respondents (91.0% of the total survey respondents). Among the 26,399 respondents, we excluded 21,124 respondents who reported that they had no child (*n* = 16,899), or that all their children were over 14 years old (*n* = 4225). Then, we excluded 19 outliers who were younger than the age at which people are allowed to be legally married, that is, 18 years old for men (*n* = 15) and 16 years old for women (*n* = 4). Furthermore, 96 respondents with two or more children of the same age were excluded because we could not assume which child the participants answered for in the questions of interest. Moreover, we excluded seven respondents who reported that they had more than five children because the maximum number of children that the respondents could answer for was five due to the survey structure as they might have failed to provide the appropriate outcome variable; we thus included 5153 respondents at this step. Lastly, we excluded 1788 respondents due to invalid primary-outcome information. As a result, we analyzed 3365 participants in the present study (Figure 1).

Among 3365 participants, 762 (22.6%) were single parents (Table 1). The proportion of female respondents was 49.2% in two-parent households and 90.0% in single-parent households. The mean parent age was younger among single-parent households. Educational attainment and household income level tended to be higher among two-parent households. The proportion of non-regular employees was higher among single-parent households. The proportion of the youngest child age group (0–5 years old) was higher among two-parent households.

During the state of emergency declaration, the proportion of childcare by others was higher among single-parent households (Table 1). While more single parents reported school and nursery closure than married parents, educational opportunities and online and cram schools were not significantly different between single-parent and two-parent children.

Overall, children in single-parent households spent a shorter time carrying out physical activities and reading, and spent a longer time studying at home, watching television and online entertainment and playing games than those in two-parent families (Table 1). On the other hand, sleeping hours were not significantly different. The distribution of the daily activities by child age group is shown in Table A1 (Appendix A).

Within two-parent households, the covariates, namely, the respondents’ age, household income level and educational attainment were significantly different according to respondents’ gender, except for the status of living with a grandparent (Table A2).

### 3.2. Children’s Emotional Instability

As the primary result in the present study, the probability of emotional instability was higher in single-parent households compared with that in two-parent households; the adjusted prevalence ratio (95% CI) was 1.26 (1.07–1.49) (Table 2).

As the results of analyses using secondary outcomes, we observed similar trends, with the main result being that children in single-parent households were more likely to use abusive language towards others, become demotivated to study and be absent from school but not perpetrate violence towards others (Table 2).

When we stratified by children’s age, among the 13–14-year-old group, the prevalence ratio of children’s emotional instability in single-parent households was statistically significantly higher than that of two-parent households (Table 3). However, we did not observe a significant interaction between children’s age and family type (*p*-interaction = 0.49, 0.74 and 0.54 in 6–9-year-old, 10–12-years-old and 13–14-year-old groups, respectively). The stratification for children’s age of the secondary outcome analyses also showed the trend of increasing prevalence ratios with the increase in the children’s age, except regarding violence towards others (Table A3). We found the interaction between family type and the children’s age of 13–14 years old to be insignificant.

### 3.3. Stratifying by Environment and Daily Activities

Table 4 shows the results of stratified analyses according to environment based on the association between children’s emotional instability and the *p*-value of the interaction terms between the family type and environment. Among those who experienced school or nursery closure, single-parent children had a higher prevalence ratio of emotional instability than two-parent children; the adjusted prevalence ratio was 1.24 (1.04–1.48). On the contrary, children who attended school or nursery did not show a significant prevalence ratio of emotional instability between two-parent and single-parent households.

Furthermore, Table 5 shows the results of stratified analyses according to children’s daily activities based on the association between children’s emotional instability and *p*-values of the interaction terms between the family type and daily activities. Among children who had shorter sleeping hours, a shorter reading time, a longer screen time and who spent a longer time playing games, children living in single-parent households had higher prevalence ratios of emotional instability than those in two-parent households.

## 4. Discussion

This study aimed to investigate the impact of family type (single-parent and two-parent households) on children’s emotional instability during the COVID-19 outbreak in Japan. The probability of emotional instability was higher in single-parent households. Additionally, changes in children’s environments due to the pandemic, such as school or nursery closure, affected children’s emotional stability disproportionately according to family type. To the best of our knowledge, this is the first cross-sectional study to investigate the impact of single-parent households on children’s mental status during the COVID-19 pandemic in Japan.

The association between family type and emotional instability was consistently evident in terms of both parent-reported emotional status and perceived misbehavior. Misbehaviors, namely, violence towards others, abusive language towards others, demotivation to study and absenteeism, reflect children’s mental health problems [17,18,19,20]. The trend was pronounced among children aged 13–14 years old (i.e., junior high school age). If this emotional instability leads to or is a symptom of depression, this trend will be a severe problem because depression is associated with devastating consequences, such as poor academic performance [21] and suicide [22,23] in this life stage. Although the present study did not investigate the severity of children’s mental status, the trajectory in children’s psychosocial conditions, especially in single-parent households, would be exacerbated if the social situation for children remained devastated by the COVID-19 pandemic.

The results of the stratified analyses suggested several considerable effect modifiers regarding the association with children’s emotional instability. For example, the adjusted prevalence ratios of emotional instability among the children who could not go to school or nursery were higher than those among the children who did. On the contrary, we observed a higher adjusted prevalence ratio of emotional instability among single-parent children both with and without online education. This result suggests that the online educational opportunities did not mitigate the risk of a deteriorating emotional status among single-parent children, even though children could communicate with their teachers and keep learning. Moreover, over 30% of the parents reported that their children had opportunities to access online education. However, children who accessed online education showed a higher prevalence of mental instability among both family types. There may be la ack of transferring school functions. Therefore, online classes might have exacerbated children’s anxiety. Of note, the impact of online education, including ERT, reported to date is heterogeneous. Even with remote learning opportunities, school closure contributes to increased anxiety and loneliness among adolescents and emotional instability among children [17]. On the other hand, there was no association between ERT and children’s risk for anxiety, depression and obsessive compulsive disorder during the six months from April 2020 to October 2020 in Florida, USA [24]. These findings support that the opening of schools and nurseries could lessen the negative effect of family type on children’s emotional instability. Schools can serve as mental health care providers for children and adolescents [18]. Without schools or nurseries, children may have lacked holistic support that would have been otherwise provided. According to a national survey, only 20% of the municipalities opened consultation counters designated for children, and 59% secured places in schools for children [10]. The lack of contact with adults other than their family or physical locations apart from their homes could have influenced children’s emotional states. Even without classes or formal education, maintaining occasional connection between teachers and children via telephone or online tools could have a beneficial effect on children’s emotional states, especially among those with less parental support or financial hardship, since they are likely to be at a high risk of mental problems [20]. Additionally, there are some public resources, such as toll-free hotlines, designated specifically for children in Japan [25,26]. These measures should have been more advertised through the educational system. Given the burden on schools, more rigorous, nationwide support for offering comprehensive care is required to safeguard children in terms of mental health in case of school closure.

Some daily activities, namely, shorter sleeping hours, a longer time watching TV and online entertainment and a longer time playing games were also considerable effect modifiers regarding the association with children’s emotional instability. Apart from household types, sleep and playing games are suggested to be associated with psychological problems among children during the COVID-19 pandemic [14,27]. Our findings suggested that children who spent more than 2 h watching TV and online entertainment showed a higher prevalence of emotional instability in both two-parent and single-parent households than children who spent less than 2 h on these activities. Additionally, single-parent households were associated with a higher prevalence ratio of emotional instability only among children who spent a longer time watching TV and online entertainment. This finding was aligned with prior reports that a longer screen time was associated with psychosocial problems [28,29].

There are some limitations of the present study. First, this was a cross-sectional study; hence, we cannot infer any causal links between household types and children’s mental status. Data on the target children’s mental health before or after the pandemic were not available. This made it challenging to assess whether children’s mental instability was associated solely with the pandemic and its related countermeasures. However, we asked parents whether their children “became” mentally instable during the declaration of a state of emergency, instead of “were,” to reduce the possibility of capturing an already worsened mental status since before the declaration. The fact that no prior research has investigated mental instability by family type in Japan might highlight the significance of this study. Second, the misclassification of children’s emotional status is a concern because this status was reported by their parents, not by children themselves. If parental reporting caused non-differential misclassification, our main findings were likely underestimated. Third, the adjustment of multivariable regression analysis was performed according to the covariates reported by only one of the two parents among two-parent households. As shown in Table A2, it is likely that fathers and mothers belonged to different age categories and educational backgrounds. Fourth, there was a limitation in the generalizability of our findings due to the study design. As this survey was commercial- and web-based, participants were likely familiar with internet use and online surveys. Participation was voluntary and compensated with a small benefit (provided via points used as cash in Rakuten service), so that they had spare time for voluntary activities. Prior studies from JACSIS adopted a weighing method to adjust responses to improve generalizability [15]; however, a weighing score was not available in the 1000 single-parent respondents in the present study. Fifth, a limited number of single fathers were recruited in this study. Only 76 out of 762 (10.0%) single parents were males among the eligible sample (*n* = 3365). Nevertheless, the proportion was not far from the national proportion of fathers among single parents in Japan, 11.2% according to the census in 2017 [30]. Hence, we believe that the gender balance in the present study reflected the current circumstances in Japan. Lastly, the JACSIS questionnaire did not ask for the children’s gender, which might affect their emotional instability. However, children’s gender was not likely to have affected their parents’ participation in this study, and therefore, we can assume that there was no significant gender disproportion among the analyzed children.

## 5. Conclusions

A large-scale cross-sectional internet survey in Japan indicated that children in single-parent households were more likely to be emotionally instable than those in two-parent households during the declaration of a state of emergency in Japan. Our findings suggest a disproportionate impact of policies on children’s health. Further investigation of the potential causes of youth health inequity is warranted, and policies that support young people under a pandemic or a state of emergency should be revisited.

## Figures and Tables

**Figure 1 ijerph-19-04239-f001:**
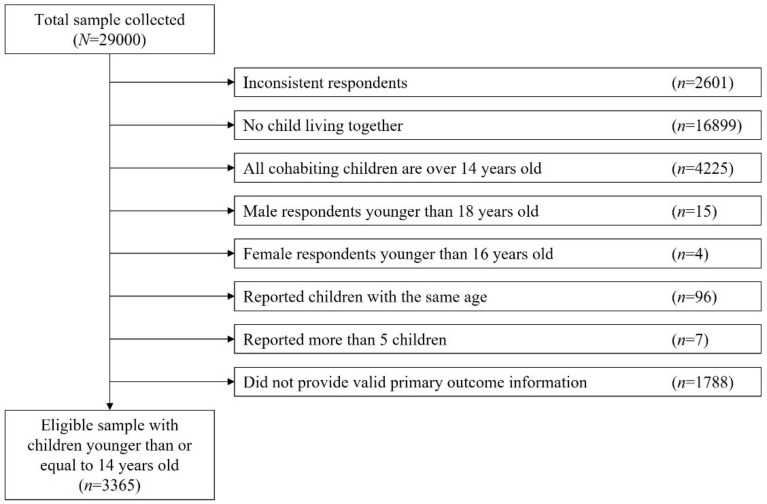
Flow chart of the study population.

**Table 1 ijerph-19-04239-t001:** Parent and child characteristics by family type (*n* = 3365).

	Family Type			
	Two-Parent	Single-Parent	Total	
	(*n* = 2603)	(*n* = 762)	(*n* = 3365)	*p*
Response from mother, %	49.2	90.0	58.5	<0.01 ^3^
Age, mean (SD)	40.7 (7.4)	39.7 (6.7)	40.5 (7.2)	<0.01 ^4^
Educational attainment, %				<0.01 ^3^
High school or lower	19.5	32.0	22.3	
College or higher	80.4	67.6	77.5	
Household income level, % ^1^				<0.01 ^3^
Higher	23.3	8.8	20.0	
Intermediate	43.9	17.2	37.8	
Lower	21.9	60.6	30.7	
Not answered	10.9	13.4	11.4	
Number of children, mean (SD)	1.85 (0.75)	1.57 (0.75)	1.79 (0.76)	<0.01 ^4^
Number of family member, % ^2^				<0.01 ^3^
≤3 persons	32.3	65.4	39.8	
4 persons	46.9	18.0	40.4	
≥5 persons	20.7	16.7	19.8	
Employment status, %				<0.01 ^3^
Employer	3.6	2.9	3.5	
Self-employed	3.8	4.3	3.9	
Regular employee	54.4	52.2	53.9	
Non-regular employee	17.1	32.8	20.7	
Unemployed	21.1	7.7	18.1	
Living with a grandparent, %	7.7	32.4	13.3	<0.01 ^3^
Children’s age group, % ^5^				<0.01 ^3^
0–5 years old	30.7	18.2	27.9	
6–9 years old	28.1	25.6	27.5	
10–12 years old	30.5	36.6	31.9	
13–14 years old	10.7	19.6	12.7	
School/nursery closure, %				0.07 ^3^
Open	12.3	13.5	12.6	
Closed	76.0	77.7	76.3	
Not answered	11.7	8.8	11.1	
Online education				0.41 ^3^
No	65.8	67.5	66.2	
Yes	31.8	30.8	31.6	
Not answered	2.4	1.7	2.3	
Childcare by others, %				<0.01 ^3^
No	20.2	27.2	21.8	
Yes	1.8	1.6	1.8	
Not answered	7.1	5.4	6.7	
Extracurricular activity participation				
cram school, %				0.56 ^3^
No	17.6	17.5	17.6	
Yes	1.6	1.0	1.5	
Not answered	6.4	4.2	5.9	
Sport/hobby lessons, %				0.37 ^3^
No	81.1	83.1	81.5	
Yes	17.9	16.3	17.6	
Not answered	1.0	0.7	0.9	
Daily activities, %				
Sleep: ≥8 h	81.9	79.7	81.4	0.32 ^3^
Study: ≥1 h	49.5	50.9	49.8	<0.01 ^3^
Physical activity: ≥30 min	53.5	40.0	50.5	<0.01 ^3^
Reading: ≥30 min	39.3	33.3	37.9	<0.01 ^3^
Watching TV/online entertainment: ≥2 h	44.4	50.8	45.9	<0.01 ^3^
Game: ≥1 h	43.0	53.1	45.3	<0.01 ^3^

Notes: SD = standard deviation. ^1^ As equivalized household income, “low” = less than 2.5 million JPY; “medium” = 2.5 to 4.3 million JPY; and “high” = more than 4.3 million JPY. ^2^ Including respondents. ^3^ We performed Chi-square tests for categorical variables. ^4^ We performed a Mann–Whitney test. ^5^ The proportion of the target child age group is shown.

**Table 2 ijerph-19-04239-t002:** Association between family type and children’s outcome (emotional instability and misbehaviors) (*n* = 3358).

		Total Number	Prevalence	Crude PR (95%CI)	Adjusted PR (95% CI) ^1^
Emotional instability					
	Two-parent	2603	17.8%	Reference	Reference
	Single-parent	762	23.1%	1.30 (1.11–1.51)	1.26 (1.07–1.49)
Violence towards others					
	Two-parent	2603	6.7%	Reference	Reference
	Single-parent	762	7.1%	1.06 (0.79–1.42)	1.01 (0.74–1.38)
Abusive language towards others					
	Two-parent	2603	14.3%	Reference	Reference
	Single-parent	762	19.0%	1.35 (1.13–1.60)	1.23 (1.02–1.49)
Demotivation to study					
	Two-parent	2603	26.2%	Reference	Reference
	Single-parent	762	35.8%	1.35 (1.21–1.51)	1.37 (1.22–1.55)
Absenteeism					
	Two-parent	2603	3.6%	Reference	Reference
	Single-parent	762	6.8%	1.83 (1.32–2.54)	1.72 (1.17–2.53)

Notes: PR = prevalence ratio; 95% CI = 95% confidence interval. ^1^ Adjusted for respondents’ age group, educational attainment, household income and whether they lived with a grandparent.

**Table 3 ijerph-19-04239-t003:** Family type and children’s emotional instability: stratified by children’s age (*n* = 3358).

Children’s Age Group		Total Number	Prevalence of Emotional Instability	Crude PR (95%CI)	Adjusted PR (95% CI) ^1^	*p*-Interaction ^2^
0–5 years old						Reference
	Two-parent	1557	6.9%	Reference	Reference	
	Single-parent	271	8.5%	1.22 (0.81–1.85)	1.13 (0.72–1.77)	
6–9 years old						0.49
	Two-parent	1028	15.1%	Reference	Reference	
	Single-parent	275	15.3%	1.02 (0.75–1.37)	0.93 (0.67–1.30)	
10–12 years old						0.74
	Two-parent	1054	14.3%	Reference	Reference	
	Single-parent	382	18.6%	1.34 (1.04–1.71)	1.20 (0.91–1.57)	
13–14 years old						0.54
	Two-parent	377	13.5%	Reference	Reference	
	Single-parent	213	18.8%	1.47 (1.02–2.11)	1.68 (1.13–2.49)	

Notes: PR = prevalence ratio; 95% CI = 95% confidence interval. ^1^ Adjusted for respondents’ age group, educational attainment, household income and whether they lived with a grandparent. ^2^ Two-parent and 0–5 years were used as the reference to assess *p* values of the interaction between family type and children’s age group.

**Table 4 ijerph-19-04239-t004:** Family type and children’s emotional instability: stratified by children’s environment and extracurricular activity.

			Total Number	Prevalence of Emotional Instability	Crude PR (95%CI)	Adjusted PR (95%CI) ^1^	*p*-Interaction ^2^
School/nursery							0.87
	Open						
		Two-parent	391	7.7%	Reference	Reference	
		Single-parent	120	10.0%	1.25 (0.66–2.35)	1.03 (0.51–2.08)	
	Closed						
		Two-parent	2518	15.6%	Reference	Reference	
		Single-parent	781	19.5%	1.29 (1.10–1.52)	1.24 (1.04–1.48)	
Online education							0.83
	No						
		Two-parent	2844	9.3%	Reference	Reference	
		Single-parent	816	12.6%	1.30 (1.06–1.59)	1.25 (1.00–1.56)	
	Yes						
		Two-parent	1027	18.2%	Reference	Reference	
		Single-parent	295	24.4%	1.36 (1.08–1.71)	1.31 (1.02–1.69)	
Childcare by others							0.42
	No						
		Two-parent	3194	9.9%	Reference	Reference	
		Single-parent	833	13.1%	1.29 (1.06–1.57)	1.28 (1.04–1.58)	
	Yes						
		Two-parent	676	20.7%	Reference	Reference	
		Single-parent	273	23.1%	1.14 (0.89–1.47)	1.03 (0.78–1.37)	
Cram school							0.89
	No						
		Two-parent	3303	10.6%	Reference	Reference	
		Single-parent	940	13.9%	1.27 (1.06–1.52)	1.27 (1.04–1.55)	
	Yes						
		Two-parent	584	19.2%	Reference	Reference	
		Single-parent	172	25.0%	1.32 (0.99–1.78)	1.14 (0.83–1.56)	
Sport/hobby lessons							0.98
	No						
		Two-parent	3338	10.4%	Reference	Reference	
		Single-parent	972	13.9%	1.30 (1.09–1.55)	1.28 (1.05–1.55)	
	Yes						
		Two-parent	576	20.0%	Reference	Reference	
		Single-parent	144	27.1%	1.28 (0.94–1.73)	1.21 (0.87–1.67)	

Notes: PR = prevalence ratio; 95% CI = 95% confidence interval. ^1^ Adjusted for respondents’ age group, educational attainment, household income and whether they lived with a grandparent. ^2^ Two-parent and open school or nursery and “No” to each activity were used as the reference of the interaction terms.

**Table 5 ijerph-19-04239-t005:** Family type and children’s emotional instability: stratified by children’s daily activities.

Emotional Instability			Total Number	Prevalence of Emotional Instability	Adjusted PR (95%CI) ^1^	*p*-Interaction ^2^
Sleep						0.07
	<8 h					
		Two-parent	422	18.2%	Reference	
		Single-parent	137	30.7%	1.66 (1.15–2.40)	
	≥8 h					
		Two-parent	2133	17.9%	Reference	
		Single-parent	607	21.4%	1.16 (0.96–1.40)	
Study						0.47
	<1 h					
		Two-parent	803	19.2%	Reference	
		Single-parent	288	25.7%	1.38 (0.98–1.94)	
	≥1 h					
		Two-parent	1289	19.2%	Reference	
		Single-parent	388	23.5%	1.21 (0.90–1.62)	
Physical activity						0.80
	<30 min					
		Two-parent	1101	20.0%	Reference	
		Single-parent	430	24.4%	1.31 (0.98–1.74)	
	≥30 min					
		Two-parent	1393	16.8%	Reference	
		Single-parent	305	21.6%	1.27 (0.91–1.78)	
Reading						0.53
	<30 min					
		Two-parent	1486	17.7%	Reference	
		Single-parent	482	22.2%	1.36 (1.03–1.78)	
	≥30 min					
		Two-parent	1022	18.8%	Reference	
		Single-parent	254	26.0%	1.36 (0.96–1.92)	
Watching TV/online entertainment						0.15
	<2 h					
		Two-parent	1364	15.2%	Reference	
		Single-parent	351	16.5%	1.11 (0.79–1.57)	
	≥2 h					
		Two-parent	1156	21.4%	Reference	
		Single-parent	387	28.9%	1.43 (1.07–1.89)	
Game						0.18
	<1 h					
		Two-parent	1396	15.8%	Reference	
		Single-parent	331	17.2%	1.08 (0.77–1.52)	
	≥1 h					
		Two-parent	1120	20.8%	Reference	
		Single-parent	405	27.9%	1.42 (1.06–1.89)	

Notes: PR = prevalence ratio; 95% CI = 95% confidence interval. ^1^ Adjusted for respondents’ age group, educational attainment, household income and whether they lived with a grandparent. ^2^ Two-parent and shorter time of each activity were used as the reference of the interaction terms.

## Data Availability

The datasets used and analyzed during the current study are available from the corresponding author on reasonable request.

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
