# Peer review of "Did Children in Single-Parent Households Have a Higher Probability of Emotional Instability during the COVID-19 Pandemic? A Nationwide Cross-Sectional Study in Japan"

_ijerph, 2022, doi:10.3390/ijerph19074239_

Round 1
Reviewer 1 Report
The study itself (in terms of themes) is very interesting and meaningful since the study tries to examine the impact of the COVID -19 and the declaration of emergency state in Japan on the children based on the different types of households. However, the result, that is, the probability of emotional instability was higher in single-parent households could be easily expected. Thus, in this situation, I guess the authors should have put more emphasis on the implications in terms of governmental and education policies in order to provide effective ways to solve this issues. Even though the study provides some suggestions and directions of further investigation in the discussion and conclusion sections, I strongly believe that more practical implications would be required in order for the authors to make this study more meaningful and significant on the issue that this manuscript deals with. This is the major concern of this manuscript.
Here are a few more comments.
- Explanations would be required for the authors to allocate the priority among target children (Lines 65-69) in the following paragraph.
The priority among target children was allocated as follows (highest to lowest in order): a child at a higher grade at primary school (10-12 years old); a child at a lower grade at primary school (6-9 years old); a child at junior high school (13-14 years old); and a child before primary school (0-5 years old).
- Figure 1 has very low resolution.
Reviewer 2 Report
I think this is one COVID-19 study that can contribute to the literature and have important practical implications. However, I have some major concerns.
(1). I understand different countries across the world have different regulations on homeschooling etc during the pandemic and more information on the local situation might be important to help the readers to get the context of the study;
(2). in the data analysis, since the exposure were single-parents, I am not quite sure for the covariates, whether both parents of the children from two-parents families were controlled for?
(3). my main concern on the study was that there was no benchmark: no data was available BEFORE or AFTER the pandemic, meaning we have no idea whether the difference between different parent statuses exists regardless of the pandemic. Therefore, the conclusion of the paper could be challenged.
Round 2
Reviewer 1 Report
All the request on the revisions were fully integrated in the revisions. Thus, I feel that this manuscript is ready for publication.
Reviewer 2 Report
I am happy with the revision.